# Mingling Foresight with Imagination: Model-Based Cooperative Multi-Agent Reinforcement Learning

**Zhiwei Xu, Dapeng Li, Bin Zhang, Yuan Zhan, Yunpeng Bai, Guoliang Fan**

Institute of Automation, Chinese Academy of Sciences

School of Artificial Intelligence, University of Chinese Academy of Sciences

`{xuzhiwei2019, lidapeng2020, zhangbin2020, zhanyuan2020, baiyuanpeng2020, guoliang.fan}@ia.ac.cn`

## Abstract

Recently, model-based agents have achieved better performance than model-free ones using the same computational budget and training time in single-agent environments. However, due to the complexity of multi-agent systems, it is tough to learn the model of the environment. The significant compounding error may hinder the learning process when model-based methods are applied to multi-agent tasks. This paper proposes an implicit model-based multi-agent reinforcement learning method based on value decomposition methods. Under this method, agents can interact with the learned virtual environment and evaluate the current state value according to imagined future states in the latent space, making agents have the foresight. Our approach can be applied to any multi-agent value decomposition method. The experimental results show that our method improves the sample efficiency in different partially observable Markov decision process domains.

## 1 Introduction

In recent years, reinforcement learning has made remarkable achievements in game AI [47, 12], robots [33], and autonomous driving [24]. It is primarily due to the progress of model-free reinforcement learning (MFRL) research. Unlike model-based reinforcement learning (MBRL), MFRL does not require dynamics of the environment and learns from the data generated by directly interacting with the environment. Therefore, although most environment models are unavailable in the real world, MFRL can master complex tasks through large-capacity value function approximators. However, MFRL requires large amounts of training data that can be costly and risky in reality, which means that the sample efficiency is the main bottleneck of the current MFRL algorithms. So in the problem that the dynamics model is known, we should prioritize using MBRL, which can even obtain an analytical solution to optimal control problems.

Multi-agent systems are more complex than single-agent tasks. The strategy of each agent changes during the learning process in multi-agent systems, which makes the environment unstable from the perspective of each agent. One method [44, 10, 38, 19, 22] is for agents to use channels to transmit information to each other, in this way to reach a consensus between agents. In addition, many current studies on multi-agent reinforcement learning have focused on the paradigm of centralized training with decentralized execution (CTDE) [29]. The agent only obtains global information during training can significantly alleviate the instability problem. The first branch [29, 11, 16] that emerged is using the Actor-Critic structure's advantages to build the centralized critic and decentralized actor framework. The value decomposition method [45, 39, 43, 30, 50, 49, 53] is also the most popular CTDE approach lately, and it has obtained excellent performance in the decentralized partially observable Markov decision process (Dec-POMDP) domains [32]. Nevertheless, most of the above methods are model-free and require access to an impractically large number of trajectories.

36th Conference on Neural Information Processing Systems (NeurIPS 2022).

Some work combines model-based and model-free methods. They usually learn environment models and solve control problems simultaneously, and have achieved good performance in some single-agent scenarios, such as MuJoCo [13] and Atari [20]. However, in multi-agent problems, the training of the world model faces great challenges because of the complexity of the environments. All current work on model-based multi-agent reinforcement learning can only be applicable for simple scenarios such as matrix games. In this paper, we propose **M**odel-**B**ased **V**alue **D**ecomposition (MBVD), a method that introduces the idea of model learning into value decomposition. When humans make decisions, they not only rely on the current state but also consider the future state obtained after several interactions with the environment following their current strategies. The phenomenon is called "long-term vision" or foresight. Enlightened by this ability of humans, we make agents have the foresight to cooperate by obtaining the aggregated latent states in the future. Finally, we evaluated the performance of MBVD in several different domains, including StarCraft II [41], Google Research Football [26] and Multi-Agent MuJoCo [37]. We clarified that MBVD has high sample efficiency, which exceeds the performance of other baselines in most scenarios. To the best of our knowledge, our study is the first attempt to apply the model-based ideas to Dec-POMDP problems.

## 2   Related Work

### 2.1   Single-Agent Model-based RL

The model-based approaches are divided into three different categories. The first branch is represented by Dyna-Q [46, 27, 20, 6, 52], the method of alternating the two processes, including learning environment models and improving policies. The focus of this idea is that the world model generates trajectories used by the training of reinforcement learning. Analytic-gradient algorithms can calculate the analytic gradient related to the reinforcement learning optimization goal through the parameterized world model, thereby directly improving the policy. This method is also called Policy Search with Backpropagation through Time [14, 8, 15, 7, 13]. The last method rolls out the learned models over multiple time steps to predict the value of states. VPN [31] plans and calculates the path with the largest accumulated reward in rollouts and goes back to the current moment to improve targets. MVE [9] uses an update method similar to n-step q-learning to calculate the target state value. However, a fundamental limitation is that the above algorithms rely heavily on the world model. So if the learned environment model is inaccurate, it will lead to the collapse of the entire learning process. Especially the last method suffers from the limitation because, in addition to learning the dynamics model of the environment, model-augmented value expansion algorithms [42, 51, 18, 1, 56, 3] also need to learn the reward function of the environment to calculate the accurate value function. It is impractical in sparse reward environments.

### 2.2   Multi-Agent Model-based RL

So far, there is relatively little work on multi-agent model-based RL. [54] obtained the sample complexity of multi-agent model-based RL through theoretical proof when the dynamics model was available. Nevertheless, it only applies to simple infinite-horizon zero-sum discounted Markov games and has not been implemented. $M^3$–UCRL [36] combines model-based RL with mean-field game theory, which can be used in cooperation problems like stylized swarm motion. However, due to the limitation of the mean-field theory, $M^3$-UCRL can only be suitable for the scenarios of a large population of agents. Both [35] and [55] construct the environment model that includes a transition function and a prediction model for the opponents' actions, and then train their policies with the opponent-wise rollouts. These two methods have been evaluated in the multi-particle environment, and the results demonstrate that they can reduce sample complexity. However, it is not feasible to build the opponent models without access to opponents' information. CPS [2] approximates a factored sparse Q-function, which is similar to the value decomposition method. Besides, CPS learns the environment model and maintains a replay buffer with priority, but it cannot solve the partially observable problems. Similarly, in the multi-agent model-based RL field, we also face compounding errors caused by imperfect learned models.

MBVD focuses on learning a standard forward dynamics model and aggregating the information contained in the environment model rollouts. Then the current state value, which is with respect to the imagined rollouts, can be obtained. We believe the imagined information can enable all agents to understand the current situation better.

# 3 Preliminaries

## 3.1 Dec-POMDP

Partially observable stochastic games (POSGs) are one of the most general games, and the Dec-POMDP [32] is an important subclass of POSGs. Formally, the Dec-POMDP is defined as a tuple $G = \langle S, U, A, P, r, Z, O, n, \gamma \rangle$. Each agent $a \in A := \{1, \dots, n\}$ selects the appropriate action $u^a \in U$ at each time step, with only access to the local observation $z^a \in Z$ provided by the observation function $O(s, a) : S \times A \rightarrow Z$, where $s \in S$ is the global state of the environment. $\boldsymbol{u} \in \boldsymbol{U} \equiv U^n$ denotes the combined action of all agents. The state transition function, commonly known as the environmental dynamics, is written as $P(s' \mid s, \boldsymbol{u}) : S \times \boldsymbol{U} \times S \rightarrow [0, 1]$. Noted that all agents in Dec-POMDPs have the same reward function: $r(s, \boldsymbol{u}) : S \times \boldsymbol{U} \rightarrow \mathbb{R}$. And $\gamma$ is the discount factor.

## 3.2 Value Decomposition

When there are multiple agents in the system, it is impossible to train each agent separately or treat all agents as one entity for joint training in cooperative multi-agent reinforcement learning. The appearance of value decomposition methods promotes the collaboration between agents and solves the credit assignment problem. In the Dec-POMDP, we make the following assumption:

$$\arg\max_{\boldsymbol{u}} Q_{tot}(\boldsymbol{\tau}, \boldsymbol{u}) = \left( \begin{array}{c} \arg\max_{u_1} Q_1 (\tau_1, u_1) \\ \vdots \\ \arg\max_{u_n} Q_n (\tau_n, u_n) \end{array} \right),$$

where $\boldsymbol{\tau} \in T^n$ represents the joint action-observation histories of all agents, $Q_{tot}$ is the global action-value function, and $Q_n$ is the individual ones. The assumption is known as the Individual-Global-Max (IGM) [43] principle, which states that a task may only be decentralized if there is consistency between the local greedy actions and global ones. Many multi-agent reinforcement learning algorithms have performed well by observing the IGM principle, like VDN [45] and QMIX [39].

## 3.3 Environment Models

The most notable feature of MBRL is that while training the policy, it also learns the world model, a parametric model used to simulate the environment. Two different models can be learned unsupervised from local observations and actions, namely auto-regressive and state-space models.

Auto-regressive models [21, 4] are intuitive and straightforward. However, there are two reasons for the high computational complexity of the auto-regressive models: calculated items of the generation process cannot be reused, and auto-regressive models need to render high-dimensional observations explicitly. On the contrary, state-space models [34, 25] first abstract the environment to find a compact latent state space $\mathcal{S}$ containing all vital information. In the multi-agent system, each $\hat{s}_t \in \mathcal{S}$ is an abstract representation of the local observations $\boldsymbol{z}_t$ of all agents. So the observation function can be expressed by $p(\boldsymbol{z}_t \mid \hat{s}_{0:t}, \boldsymbol{u}_{0:t-1}) = p(\boldsymbol{z}_t \mid \hat{s}_t)$. Using imagined rollouts obtained by the transition function on the low-dimensional latent state space, state-space models can significantly reduce the amount of calculation. The factorization of the predictive distribution is as follows:

$$p(\boldsymbol{z}_{1:T}, \boldsymbol{u}_{0:T}) = p_{\text{init}}(\hat{s}_0) \int \prod_{t=1}^{T} (p (\hat{s}_t \mid \hat{s}_{t-1}, \boldsymbol{u}_{t-1}, \boldsymbol{z}_t) \, p(\boldsymbol{u}_t \mid \boldsymbol{z}_t) p(\boldsymbol{z}_t \mid \hat{s}_t)) \, d\hat{s}_{1:T},$$

where $p_{\text{init}}(\hat{s}_0)$ denotes the prior distribution of the initial state $\hat{s}_0$, which is usually a constant. $p(\hat{s}_t \mid \hat{s}_{t-1}, \boldsymbol{u}_{t-1}, \boldsymbol{z}_t)$ is the true posterior of the latent state, and $p(\boldsymbol{u}_t \mid \boldsymbol{z}_t)$ means the joint policy $\boldsymbol{\pi}$ of all agents.

# 4 Model-Based Value Decomposition

This section will elaborate on MBVD, a novel model-based multi-agent reinforcement learning algorithm based on value decomposition. We will introduce the framework and flowchart of MBVD first and then describe the detailed implementation of MBVD.

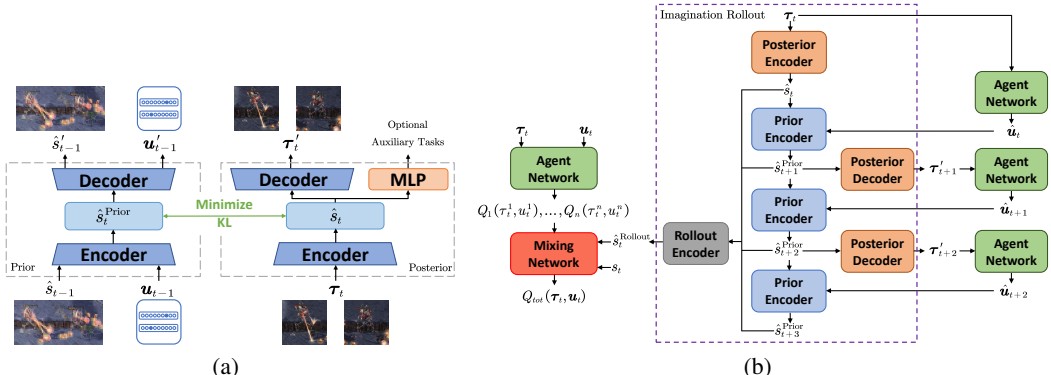

Figure 1: Illustration of MBVD implementation. (a) The imagination module in MBVD. (b) The workflow of MBVD. The rollout horizon in the figure is 3.

## 4.1 The MBVD Framework

To maintain the applicability of algorithms, the framework of the reinforcement learning part in MBVD is consistent with other value decomposition methods, including the agent network and the mixing network. We focus on the model learning part of MBVD, called the imagination module.

Inspired by amortized variational inference, we employ neural networks to approximate the intractable posterior $p(\hat{s}_t \mid \hat{s}_{t-1}, \boldsymbol{u}_{t-1}, \boldsymbol{z}_t)$. The approximate posterior is defined by $q_\theta(\hat{s}_t \mid \hat{s}_{t-1}, \boldsymbol{u}_{t-1}, \boldsymbol{z}_t)$, where $\theta$ is the parameters. So we can derive the evidence lower bound (ELBO) as follows:

$$\log p\left(\boldsymbol{z}_{1:T}, \boldsymbol{u}_{0:T}\right)$$
$$= \log \mathbb{E}_{q_\theta(\hat{s}_{1:T}\mid\boldsymbol{u}_{0:T},\boldsymbol{z}_{1:T})} \left[ \frac{p\left(\hat{s}_{1:T}, \boldsymbol{u}_{0:T}, \boldsymbol{z}_{1:T}\right)}{q_\theta\left(\hat{s}_{1:T} \mid \boldsymbol{u}_{0:T}, \boldsymbol{z}_{1:T}\right)} \right]$$
$$\geq \mathbb{E}_{q_\theta(\hat{s}_{1:T}\mid\boldsymbol{u}_{0:T},\boldsymbol{z}_{1:T})} \log \left[ \frac{p\left(\hat{s}_{1:T}, \boldsymbol{u}_{0:T}, \boldsymbol{z}_{1:T}\right)}{q_\theta\left(\hat{s}_{1:T} \mid \boldsymbol{u}_{0:T}, \boldsymbol{z}_{1:T}\right)} \right],$$

and the ELBO can be broken down as:

$$\mathcal{L}\left(\boldsymbol{z}_{1:T}, \boldsymbol{u}_{0:T}\right)$$
$$= \sum_{t=1}^{T} \left\{ \log\left[p(\boldsymbol{u}_t \mid \boldsymbol{z}_t)\right] + \log\left[p(\boldsymbol{z}_t \mid \hat{s}_t)\right] - \mathcal{D}_{\mathrm{KL}}\left[q_\theta\left(\hat{s}_t \mid \hat{s}_{t-1}, \boldsymbol{u}_{t-1}, \boldsymbol{z}_t\right) \| p\left(\hat{s}_t \mid \hat{s}_{t-1}, \boldsymbol{u}_{t-1}\right)\right] \right\}. \tag{1}$$

The first term $\log\left[p(\boldsymbol{u}_t \mid \boldsymbol{z}_t)\right]$ is the joint policy and we can ignore it. Furthermore, the term $\log\left[p(\boldsymbol{z}_t \mid \hat{s}_t)\right]$ can be viewed as the observation model. In addition, we propose a new parameterized function $p_\phi^{\mathrm{Prior}}(\hat{s}_t \mid \hat{s}_{t-1}, \boldsymbol{u}_{t-1})$ to approximate the dynamics model $p(\hat{s}_t \mid \hat{s}_{t-1}, \boldsymbol{u}_{t-1})$, which is unavailable.

MBVD follows the idea of planning in latent spaces, so we use the Variational Autoencoder (VAE) [23] to maximize the ELBO. The reason is that generative models can find a low-dimensional space by reconstructing the original data, which is consistent with abstracting the decision-related state from the real observations. For the last term in Equation 1, we regard the two items in Kullback-Leibler (KL) divergence as the posterior and the prior. Intuitively, according to Equation 1, we need to minimize the KL divergence between the posterior $q_\theta(\cdot)$, which incorporates information about the current observations $\boldsymbol{z}$, with the prior $p_\phi^{\mathrm{Prior}}(\cdot)$ that tries to predict the posterior without access to the current observations. Then we use two VAEs as the prior and the posterior, respectively, as shown in Figure 1(a). In this way, we obtain the prior latent state and the posterior latent state:

$$\hat{s}_t^{\mathrm{Prior}} \sim p_\phi^{\mathrm{Prior}}\left(\cdot \mid \hat{s}_{t-1}, \boldsymbol{u}_{t-1}\right),$$
$$\hat{s}_t \sim q_\theta\left(\cdot \mid \hat{s}_{t-1}, \boldsymbol{u}_{t-1}, \boldsymbol{z}_t\right).$$

However, we use a modified version of the posterior model in the implementation process. Since the hidden output $h_t^a$ of the recurrent neural network in the agent network can be regarded as the

integration of all past information of the individual agent, the posterior $q_\theta(\hat{s}_t \mid \hat{s}_{t-1}, \boldsymbol{u}_{t-1}, \boldsymbol{z}_t)$ can be rewritten as $q_\theta(\hat{s}_t \mid \boldsymbol{h}_t)$. The advantages of minimizing the KL loss are that we can not only guide the prior to predicting the latent posterior state distribution without the information of the current step but also control how much information the posterior abstracts from the original observation. It is worth noting that the learned prior model can predict forward in the latent space based on the actions of all agents, even if the actions are different from reality. So the imagination module in MBVD we proposed is an action-conditional environment model. From another perspective, we can view the prior model as the transition function, and the decoder $q_\theta(\boldsymbol{z}_t \mid \hat{s}_t)$ of the posterior can be regarded as the observation function. Through the above framework, we have built the imagination module that is general and model-agnostic for the multi-agent value decomposition methods.

In addition to learning the transition function and observation function, in some complex environments such as StarCraft II, the imagination module also needs to predict the feasible action set $\mathcal{A}$ to choose imagined actions more reasonably. Since the posterior can access the actual information of the current step, we can perform multiple auxiliary tasks by using the latent state $\hat{s}_t \sim q_\theta(\hat{s}_t \mid \boldsymbol{h}_t)$ as the intermediate variable to predict the additional environmental signals. The reward function $r(\cdot)$ is crucial in reinforcement learning because it steers the agents' behavior. However, predicting the reward function in noisy, sparse-reward, or complex environments is often difficult. So the imagination module in MBVD does not utilize the reward signal in the environment. MBVD adopts the implicit model-based idea without reward prediction.

The imagination model we offered has all the essential components of the environment, except the action-value model, which can evaluate the expected returns. We leave the implementation of the action-value model to the reinforcement learning process. With the help of the imagined states generated from the simulated model, the agents seem to have the foresight, which means that the agents estimate the action value based on long-term consequences rather than direct results. Thus, we can discern the role of the imagination module in MBVD.

## 4.2   The MBVD Flow Diagram

The generation process of imagined rollouts can be found in Figure 1(b). At time $t$, $\boldsymbol{\tau}_t$ refers to the trajectory history of all agents. Since the posterior encoder can be regarded as the inverse function of the observation function, we can infer the latent state $\hat{s}_t$ from $\boldsymbol{\tau}_t$. Furthermore, according to the current joint policy $\boldsymbol{\pi}$ of all agents (implemented by the agent network), we can get the actions $\hat{\boldsymbol{u}}_t \sim \boldsymbol{\pi}(\cdot \mid \boldsymbol{\tau}_t)$ that all agents should perform to maximize the estimate of the state-action value. Then the learned prior model is used to derive the next latent state $\hat{s}_{t+1}^{\text{Prior}}$ based on $\hat{s}_t$ and $\hat{\boldsymbol{u}}_t$. To get the joint action $\hat{\boldsymbol{u}}_{t+1}$ at the next step in the imagined rollouts, we need to obtain the trajectory history $\boldsymbol{\tau}'_{t+1} \sim q_\theta\left(\cdot \mid \hat{s}_{t+1}^{\text{Prior}}\right)$ corresponding to the latent state $\hat{s}_{t+1}$ through the observation function. We perform one step forward in the imagined rollout in the above way. Then we can roll out the environment model over multiple time steps by feeding the following imagined variables into the model. As mentioned above, we use all individual hidden outputs $\boldsymbol{h}$ instead of the trajectories $\boldsymbol{\tau}$ to accelerate the training speed in the implementation.

We use the latest policies of all agents as the rollout policies. Here is a reasonable and intuitive explanation: when people make a strategic decision, they always use their current policies as the imagined policies for their decision-making, rather than previous strategies that have led to poor performance. Besides, for the stability of reinforcement learning, we turn the random process involved in the imagination module during reinforcement learning into a deterministic form. For example, the imagined actions $\hat{\boldsymbol{u}}$ are selected from greedy policies. Moreover, we directly take the mean of the approximating distribution as the inferred latent state $\hat{s}$ and trajectory history $\hat{\boldsymbol{\tau}}$. However, we still use the reparameterization trick when learning the imagination module.

One $k$-step rollout starting with $\hat{s}_t$ can be written as $\left\{ (\hat{s}_t, \boldsymbol{\tau}_t, \hat{\boldsymbol{u}}_t), (\hat{s}_{t+1}^{\text{Prior}}, \boldsymbol{\tau}'_{t+1}, \hat{\boldsymbol{u}}_{t+1}), \ldots, (\hat{s}_{t+k}^{\text{Prior}}, \boldsymbol{\tau}'_{t+k}, \hat{\boldsymbol{u}}_{t+k}) \right\}$. Since the latent state $\hat{s}$ is inferred from all agents' historical trajectories and actions, $\hat{s}$ contains the information of $\boldsymbol{\tau}$ and $\boldsymbol{u}$. Furthermore, to ensure that the parameters of the imagination module do not increase with the rollout horizon $k$, the set of imagined latent states $\{\hat{s}_t, \hat{s}_{t+1}^{\text{Prior}}, \ldots, \hat{s}_{t+k}^{\text{Prior}}\}$ is fed into the GRU [5] to get the aggregated rollout state $\hat{s}_t^{\text{Rollout}}$. Finally, we concatenate the aggregated rollout state $\hat{s}_t^{\text{Rollout}}$ with the real global state $s_t$ and input them into the mixing network of the value decomposition framework. We believe that the imagined states contain information about the possible states of the future, which can help all agents evaluate the current state more accurately.

MBVD does not require a pre-trained environment model, which means the simulated model and the action-value model are all trained from scratch simultaneously. As an end-to-end efficient MBRL algorithm, MBVD can be extended to any value decomposition method with the mixing network.

## 4.3 Overall Learning Objective

Next, we will elaborate on the training objectives of MBVD. MBVD involves two processes: the reinforcement learning process, whose objective is to minimize the td-error; the other is learning the imagination module, which includes the optimization of the prior and the posterior. These two processes are carried out simultaneously. For the convenience of description, we define the parameters of the value decomposition framework as $\psi$.

MBVD can be seen as the value decomposition method with an imagination module, so the loss function for reinforcement learning is consistent with the original value decomposition method. The most obvious difference is that the global action-value function $Q_{tot}$ is calculated with respect to the actual state $s$ and the latent rollout state $\hat{s}^{\text{Rollout}}$. It is important to note that MBVD generates the aggregated rollout state $\hat{s}^{\text{Rollout}}$ conditioned on the past state $s$ in the replay buffer, so we do not change the off-policy update paradigm. The loss function for reinforcement learning can be obtained:

$$\mathcal{L}_{\text{RL}} = \left( y^{tot} - Q_{tot}(\boldsymbol{\tau}_t, \boldsymbol{u}_t, s_t, \hat{s}_t^{\text{Rollout}}; \psi) \right)^2,$$

where $y^{tot} = r_t + \gamma \max_{\boldsymbol{u}_{t+1}} Q_{tot} \left( \boldsymbol{\tau}_{t+1}, \boldsymbol{u}_{t+1}, s_{t+1}, \hat{s}_{t+1}^{\text{Rollout}};\ \psi^- \right)$, and $\psi^-$ represents the parameters of the target network.

The loss function of the posterior can be divided into reconstruction loss and KL divergence loss. The posterior model infers the current latent state after the observations of all agents are given, which requires that the posterior model extract helpful information from the original input. It can be achieved by narrowing the difference between the model output $\boldsymbol{\tau}'$ and the model input $\boldsymbol{\tau}$. The reconstruction loss function of the prior model is similar, and both of them can be computed by:

$$\mathcal{L}_{\text{RC}} = \text{MSE}\left(\boldsymbol{\tau}_t, \boldsymbol{\tau}_t'; \theta\right), \qquad \mathcal{L}_{\text{RC}}^{\text{Prior}} = \text{MSE}\left( (\hat{s}_{t-1}, \boldsymbol{u}_{t-1}), (\hat{s}_{t-1}', \boldsymbol{u}_{t-1}');\phi \right),$$

where MSE means the mean square error. Furthermore, in addition to the KL term in Equation 1, we throw in the KL divergence between the prior distribution and the standard Gaussian distribution $\mathcal{N}(0, 1)$ as a regular term to avoid the overly complex prior distribution of the latent state $\hat{s}^{\text{Prior}}$. So the KL loss is as follows:

$$\mathcal{L}_{\text{KL}} = \mathcal{D}_{\text{KL}} \left[ p_\phi^{\text{Prior}} \left( \hat{s}_t \mid \hat{s}_{t-1}, \boldsymbol{u}_{t-1} \right) \| \mathcal{N}(0, 1) \right]$$
$$+ \mathcal{D}_{\text{KL}} \left[ q_\theta \left( \hat{s}_t \mid \hat{s}_{t-1}, \boldsymbol{u}_{t-1}, \boldsymbol{z}_t \right) \| p_\phi^{\text{Prior}} \left( \hat{s}_t \mid \hat{s}_{t-1}, \boldsymbol{u}_{t-1} \right) \right].$$

We use the KL balancing mechanism to optimize the KL term in the actual implementation. KL balancing enables the prior to minimize KL loss at a faster learning rate than the posterior, which can be expressed as:

$$\mathcal{D}_{\text{KLbalancing}} \left[ q_\theta \left( \cdot \right) \| p_\phi^{\text{Prior}} \left( \cdot \right) \right] =$$
$$\alpha \mathcal{D}_{\text{KL}} \left[ q_\theta \left( \cdot \right) \| \text{Detach} \left( p_\phi^{\text{Prior}} \left( \cdot \right) \right) \right] + (1 - \alpha) \mathcal{D}_{\text{KL}} \left[ \text{Detach} \left( q_\theta \left( \cdot \right) \right) \| p_\phi^{\text{Prior}} \left( \cdot \right) \right],$$

where $\alpha \in [0, 1]$ is a configurable parameter, $\text{Detach}(\cdot)$ can stop the backpropagation from gradients of certain variables or functions.

In addition, we have other optional training objectives for auxiliary tasks in complex scenarios. In this paper, we make predictions of the feasible action set only in StarCraft II:

$$\mathcal{L}_{\text{FA}} = \text{BCE}\left(\mathcal{A}_t, \mathcal{A}_t'; \phi\right),$$

where BCE denotes the binary cross-entropy error. Thus, the total loss function can be written as:

$$\mathcal{L} = \mathcal{L}_{\text{RL}} + \mathcal{L}_{\text{RC}} + \mathcal{L}_{\text{RC}}^{\text{Prior}} + \mathcal{L}_{\text{KL}} + \mathcal{L}_{\text{FA}}. \tag{2}$$

By minimizing the total loss function $\mathcal{L}$, we can guide MBVD to accelerate reinforcement learning with the help of the imagination module.

# 5 Experiments

This section will consider whether the value decomposition method can benefit from the implicit model-based method. We compare the performance of MBVD with other popular baselines, including VDN, QMIX, MAVEN [30], Weighted QMIX [40], ROMA [49], and RODE [50]. Then by conducting the ablation experiments, we can clarify that every component in the learned model is indispensable, and different horizons of the rollout have a significant impact on performance. Finally, we will visualize the imagined rollouts, which will more intuitively and powerfully show the foresight of MBVD. Note that the implementation of agents in MBVD in all experiments is based on QMIX. The details of all experiments can be found in Appendix **??**.



| (a) StarCraft II | (b) Google Research Football | (c) Multi-Agent Discrete MuJoCo |

Figure 2: Examples of the three experimental platforms.

## 5.1 Performance on StarCraft II

SMAC is a multi-agent micro-management experimental platform based on the real-time strategy game StarCraft II, which contains a wealth of scenarios corresponding to different challenges. To verify the performance improvement brought by the imagination module, we pay more attention to the performance comparison between MBVD and QMIX. We select some representative scenarios. The median performance and the 25-75% percentiles are illustrated in Figure 3.

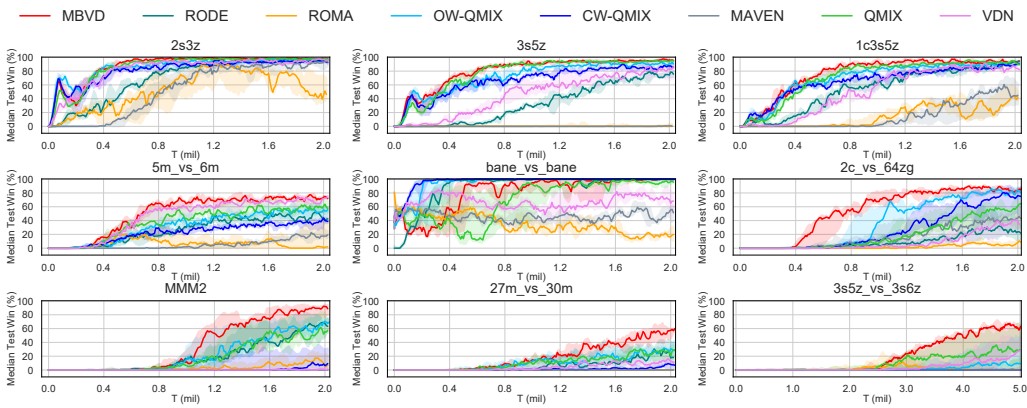

Figure 3: Performance comparison with baselines in different SMAC scenarios.

MBVD reaches state-of-the-art performance in most scenarios and exceeds the basic algorithm QMIX. Especially in hard or super hard scenarios such as *2c_vs_64zg*, *MMM2*, and *3s5z_vs_3s6z*, the superiority of MBVD is more significant. In *5m_vs_6m*, due to the uncertainty of the environment, other baselines will encounter a performance bottleneck. However, MBVD can predict the future state to achieve the best performance in *5m_vs_6m*. Even in some easy scenarios, MBVD still has high sampling efficiency. It is impossible for some complex QMIX-style variants. MBVD can robustly reduce the sampling complexity in both easy and hard scenarios.

## 5.2 Performance on Google Research Football

The Google Research Football environment provides a novel reinforcement learning environment where multiple agents can be trained to play football. To verify the effectiveness of our proposed method, we carry out MBVD and other baselines on the Football Academy, which is a diverse set of mini scenarios of varying difficulty. We select three representative official scenarios, and the experimental results are delivered in Figure 4.

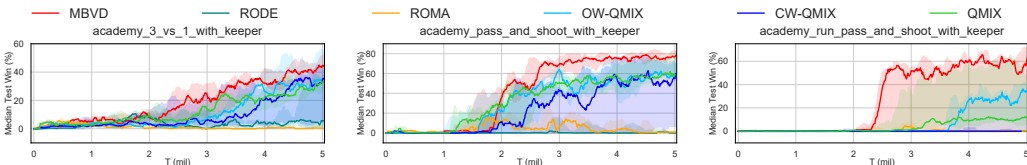

Figure 4: Performance comparison with baselines in different Google Research Football scenarios.

Compared with QMIX, which has a relatively simple structure, algorithms such as RODE and ROMA do not perform well. A possible reason is that the number of agents in these scenarios is not large, and the role assignments may hinder the learning. Conversely, MBVD based on QMIX achieves the highest sample efficiency in three scenarios with different difficulty levels. The experimental results obtained in Google Research Football also fully demonstrate the generalization capability of MBVD.

## 5.3 Performance on Multi-Agent Discrete MuJoCo

Many model-based reinforcement learning algorithms, including the work mentioned above, tend to be compared on MuJoCo. [37] recently proposed a novel benchmark for continuous cooperative multi-agent robotic control in the multi-agent field, called Multi-Agent MuJoCo. A given single robotic agent is viewed as a body graph containing many disjoint sub-graphs, and each sub-graph contains one or more joints that can be controlled. In this paper, each joint is regarded as an agent that makes local decisions conditioned on partial observations. Since this paper focuses on the impacts of the world model, we propose a discrete variant of Multi-Agent MuJoCo. Detailed environment settings can be found in Appendix **??**. We selected four representative scenarios, and Figure 5 describes the episode return of each algorithm. Since most of the relatively complex algorithms cannot converge in this environment, we only choose MBVD, QMIX, and VDN for comparison.

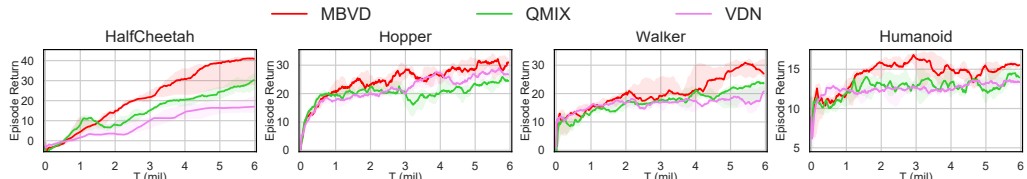

Figure 5: Median episode return on different Multi-Agent Discrete MuJoCo tasks.

From the results, we can intuitively see the performance improvement brought by the imagination module because we see MBVD as a variant of QMIX that has an additional implicit foresight module. MBVD outperformed QMIX in all given scenarios, which also means MBVD we proposed can be applied to different environments.

## 5.4 Ablation Studies

We carry out ablation studies to test the contribution of the components of the imagination module and investigate how the rollout horizon $k$ affects the performance of MBVD. For the first study, we proposed two variants of QMIX that input the additional information to the mixing network, QMIX-RS and QMIX-LS. Both of them aggregate the information of the next $k$ steps from the actual trajectories rather than imagined rollouts. However, the difference is that QMIX-RS uses the real states and QMIX-LS the latent states. To answer the second problem, we run MBVD with different rollout horizons on the *2c_vs_64zg* scenario of SMAC and compare their performance.

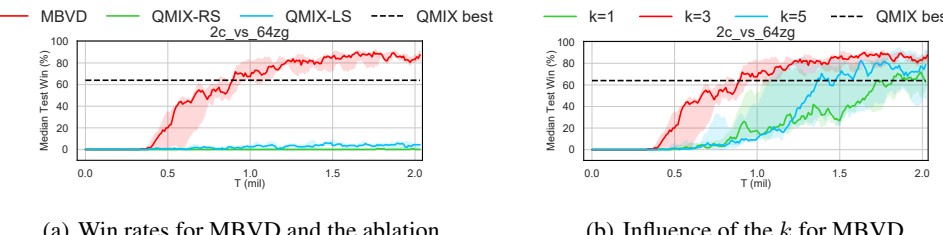

(a) Win rates for MBVD and the ablation.

(b) Influence of the $k$ for MBVD.

Figure 6: Results for ablation studies on *2c_vs_64zg* map.

As shown in Figure 6(a), both QMIX-RS and QMIX-LS failed to solve the task, which means that the transition function that can generate the imagined rollouts is critical. Besides, we can also infer the contribution of the observation function from that the better performance of QMIX-LS than QMIX-RS. For the horizon of the rollouts, we can conclude from Figure 6(b). When using short-horizon rollouts, the agents cannot fully interact with the imagined model, resulting in the same performance as vanilla QMIX. However, as $k$ increases significantly, MBVD loses the monotonic improvements because of the compounding error. The effect of $k$ holds the same for the other scenarios, but the optimal choice of $k$ in each task is different. If not explicitly stated, we will set $k$ to 3 in this paper for convenience.

## 5.5 Visualization

To intuitively explain agents' foresight ability in MBVD, we visualized the latent state sequence generated by the interaction between agents and the imagined model. For the trajectories of an episode in the *2c_vs_64zg* scenario, we visualized the t-sne embeddings of the latent states predicted at each step in the rollout and compared them with the real latent state embedding. From Figure 7, we find that the difference between the embedding of the imagined latent states and real ones gradually increases as the horizon length grows, but there are similarities between them. It also reveals that our proposed imagination module can capture and learn the dynamics of the multi-agent environment.

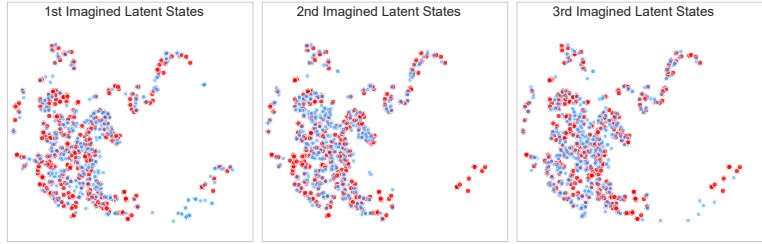

Figure 7: The 2D t-SNE embedding of real latent states (red) and imagined latent states (blue).

## 6  Discussion on the Rollout Horizon

As we describe in ablation studies, the choice of horizon length $k$ affects the performance of MBVD. Short horizons are easier to be predicted but do not provide enough information to be useful for decision-making. Long horizons can carry more valuable information, but the generated imagined rollouts are inaccurate because of compounding errors. There has been some work on model-based reinforcement learning that attempts to break this trade-off. For bias that might exist in sampling and environmental models, MBPO [17] suggests a branched rollout. And MBPO can determine the longest tolerable rollout length according to the lower bound of return. BMPO [28] uses the newly introduced bidirectional models to significantly reduce model compounding error. In addition, most of the model-based value expansion (MVE) methods [9] involve adaptive selection of horizons. They must use a dynamics model to simulate the short-term horizon and Q-learning to estimate the long-term value beyond the simulation horizon. The automatic selection of $k$ would be made possible by explicitly estimating uncertainty in the dynamics model or ensemble models. STEVE [3] suggests

interpolating the estimated values of various rollout steps, and the weight for each rollout step is chosen by considering the ensemble predictions' variance. AdaMVE [51] mitigates the detrimental effects of the compounding error by selecting the rollout horizon for any state based on the learned model error function. RAVE [56] uses probabilistic models to capture uncertainty (including aleatoric and epistemic uncertainty) and uses the lower confidence bound for value estimation to avoid optimistic estimation. DMVE [48] exploits the fact that the uncertainty of the model and the novelty of data are highly relevant in deep learning. So DMVE selects horizons based on the top $k$ minimum reconstruction errors of the auto-encoder. To sum up, in order to further solve the problem of adaptive horizon selection, we can try to select an appropriate $k$ value with the uncertainty of the imagined state as a reference.

## 7    Conclusion

Due to the high complexity of multi-agent systems, MBRL algorithms are difficult to be applied to them. This paper proposes MBVD, a novel and implicit model-based cooperative multi-agent reinforcement learning method. By learning the world model and imagining the future latent states after making several decisions under the current policy to estimate the current state value, agents in MBVD obtain the foresight ability. Through experiments and visualization, we have proved the efficiency and generalization of MBVD. This is the first study on Dec-POMDPs from the perspective of model-based reinforcement learning.

The defect in this study is that the rollout horizon is manually chosen. In our future research, we intend to concentrate on how to choose the appropriate rollout horizon. The issue is an intriguing one which could be usefully explored in further research.

## Acknowledgments and Disclosure of Funding

The work is supported by the National Defence Foundation Reinforcement Fund.

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
