# OpenReview forum: "Mingling Foresight with Imagination: Model-Based Cooperative Multi-Agent Reinforcement Learning"
_NeurIPS.cc/2022/Conference — NeurIPS 2022 Accept_

### Official Review · Reviewer_UNCB · 2022-07-09

**Rating:** 5
**Confidence:** 3
**Soundness:** 2 fair
**Presentation:** 3 good
**Contribution:** 2 fair

**Summary:**

This paper introduces the model-based value decomposition (MBVD) framework, which extends a value decomposition method (e.g., QMIX) with the imagination module. Specifically, the imagination module consists of two VAEs and outputs the rollout of imagined future states in the latent space ($\hat{s}^{\text{Rollout}}_{t}$). Then, this rollout information is concatenated with the current state ($s_t$) and becomes an input to the mixing network of QMIX. MBVD assumes that these imagined states contain helpful information for agents to evaluate the current state's value more accurately. Empirical results show that MBVD outperforms baselines in various domains of SMAC, Football, and multi-agent MuJoCo.

**Questions:**

1. In Section 4.2, the paper notes that the aggregated rollout state ($\hat{s}^{\text{Rollout}}_{t}$) is concatenated with the real global state ($s_t$). However, this paper assumes Dec-POMDP settings, so it is not possible to observe the real global state information. As such, it is confusing how the real global state information ($s_t$) can be obtained in MBVD.
2. After reading Section 5.4 (ablation studies), it is confusing to interpret the results for MBVD vs QMIX-RS. Because QMIX-RS receives the actual trajectories instead of predictions (which can be noisy approximations of true trajectories), I thought QMIX-RS would be the upper-bar performance of MBVD, but this is not true in Figure 6(a). Would it be possible to ask for a clarification?
3. Figure 7 is not straightforward to compare the real and imagined latent states. Possibly, it might be better to have 3 subfigures instead of 4 subfigures by overlaying the real latent state subfigure to the imagined latent state subfigures.
4. It is not specified how $p(u_t|z_t)$ is represented (e.g., $p(u_t|z_t)=\prod_a p(u^a_t|z^a_t)$)
5. There is a typo in Section 2.2: "MBVD we proposed".
6. A citation is missing for Dec-POMDP in Section 3.1.
7. In Section 4.2, the (fixed) latest policies of all agents are used for the rollout. However, in practice in the future, they will have non-stationary policies due to their policy updates, so I wonder whether it will be helpful to consider their non-stationary policies in generating the rollout as a future direction (related works: Foerster et al., 2018; Kim et al., 2021).

**References:**
Foerster et al., DiCE: The Infinitely Differentiable Monte-Carlo Estimator, 2018.
Kim et al., A Policy Gradient Algorithm for Learning to Learn in Multiagent Reinforcement Learning, 2021.

**Strengths And Weaknesses:**

**Strengths:**
1. The paper introduces a new framework of how learned models can be integrated into an existing value decomposition method.
2. The evaluation is performed across complex domains, demonstrating the scalability of MBVD.

**Weaknesses:**
1. Because this paper's main contribution is related to the model-based aspects, it would be desirable to have comparisons against existing model-based multi-agent baselines, but there are no comparisons against model-based multi-agent baselines in the evaluation.
2. MBVD uses the existing methods in Section 4.3 (e.g., VAE loss, QMIX loss), so there might not be a new contribution from the optimization perspective. However, as noted in Strength #1, how learned models can be combined in QMIX is new, and if this paper studied deeper on various ways to integrate learned models (other than concatenating the imagined states to the current state), this contribution could have been more significant.

---

> ### Author Response · Authors · 2022-08-01
> **We thank the reviewer for the insightful review.**
>
> We thank the reviewer for the insightful review.
>
> **Q1**: "There are no comparisons against model-based multi-agent baselines in the evaluation".
>
> **A1**: To the best of our knowledge, MBVD is the first model-based multi-agent reinforcement learning algorithm that can be perfectly applied to various Dec-POMDP tasks. None of the model-based multi-agent reinforcement learning algorithms we introduced in Section 2.2 can be directly run on the experimental platforms used in the paper.
>
> **Q2**: "It is confusing how the real global state information $s_t$ can be obtained in Dec-POMDPs".
>
> **A2**: In almost all work focused on solving the Dec-POMDP problem, the global state $s_t$ is accessible during the centralized training but not during the decentralized execution. Of course, there are also cases where the global state cannot be obtained. In this case, the local observations of all agents are generally concatenated together as the global state $s_t$.  One of the reasons that the input of the imagination module in MBVD is the local observation of agents rather than the global state is to relax the assumption that the global state is accessible. MBVD can still be applied to various value decomposition methods when the global state is unavailable.
>
> **Q3**: "QMIX-RS would be the upper-bar performance of MBVD, but this is not true in Figure 6(a)".
>
> **A3**: The actual trajectories used in QMIX-RS refer to the trajectories obtained by the agent interacting with the environment under a certain policy $\hat{\pi}$ and stored in the replay buffer. The agent's policy is updated frequently and we assume that it is $\pi$ at this time. The state sequence obtained by interacting with the environment according to the policy $\pi$ is different from that obtained by the previous policy $\hat{\pi}$. The mismatch results in poor performance of QMIX-RS as well as QMIX-LS. This is why the agent in MBVD uses the latest policy to generate imagined states. We provide new experimental results on other SMAC scenarios in Appendix C.2.
>
> **Q4**: "It is not specified how $p(\boldsymbol{u}_t|\boldsymbol{z}_t)$ is represented".
>
> **A4**: The term $p (\boldsymbol{u}_t|\boldsymbol{z}_t) $ can be expressed as $\prod_a p (u_t^a|z_t^a) $ under the assumption that observations are conditionally independent given the state. However, how $p (\boldsymbol{u}_t|\boldsymbol{z}_t) $ is represented is not important because we can ignore it in Equation 2.
>
>
>
> The imagination module in MBVD uses the double VAE structure, which is simple yet effective, and novel in model-based reinforcement learning. Furthermore, we believe that work on non-stationary policies is helpful not only in generating imagined states but also in the real policy update in multi-agent systems. Thanks again for your comments!

---

> > ### Comment · Reviewer_UNCB · 2022-08-05
> > **Response to Author Rebuttal**
> >
> > I appreciate the authors for the detailed response. I hope to follow up on A1 and A2.
> >
> > **Regarding A1:** I agree that there are small technical issues with the frameworks in Section 2.2 that make them not straightforward to apply to the considered Dec-POMDP evaluations. I still think the paper's significance could have been stronger if there were any model-based RL/MARL baselines (e.g., applying a single-agent model-based baseline to a multi-agent game), but I understand that there is not enough time to add baselines during the rebuttal.
> >
> > **Regarding A2:** The underlying assumption in the statement is that the concatenation of all agents' local observations can represent the global state. However, I am unsure whether this assumption generalizes to all possible multi-agent scenarios. As such, this is minor, but re-writing the paper based on only observations may help the technical soundness.
> >
> > The authors' responses answer my Q3 and Q4.

---

> > > ### Author Response · Authors · 2022-08-06
> > > **Further explanation of A1 and A2**
> > >
> > > We are very grateful to the reviewer for the responses! And we provide further explanation of A1 and A2.
> > >
> > > **A1**: We agree that adding single-agent model-based baselines will make the paper's contribution more significant, even though we may not be able to incorporate the results in time for the next version of our paper. Intuitively, single-agent model-based algorithms should perform poorly in multi-agent environments because of the notorious instability in multi-agent reinforcement learning. Each agent's policy is changing as training progresses, and the environment becomes non-stationary from the perspective of any individual agent. Therefore it is almost impossible for an agent to learn an accurate environmental model alone in a multi-agent system.
> > >
> > > **A2**: Almost all of the work devoted to the Dec-POMDP problem assumes that although execution is decentralized, training is centralized. In other words, each agent’s learnt policy can condition only on its own action-observation history $\tau_a$, but the learning algorithm has access to all local action-observation histories $\boldsymbol{\tau}$ and global state $s$. For example, in the original paper of the popular value decomposition method QMIX [1], it said "*QMIX relies on a neural network to transform the **centralized state** into the weights of another neural network*". Eq.(2), Eq.(6) and Figure 2(b) in [1] all illustrate that the state $s$ is available during training. Similar expressions can be found in the original paper of SMAC [2] (the environment used in our paper) or in many other studies [3, 4].
> > >
> > > **references**:
> > >
> > > [1] Rashid, Tabish, et al. Qmix: Monotonic value function factorisation for deep multi-agent reinforcement learning, 2018.
> > >
> > > [2] Samvelyan, Mikayel, et al. The starcraft multi-agent challenge, 2019.
> > >
> > > [3] Mahajan, Anuj, et al. Maven: Multi-agent variational exploration, 2019.
> > >
> > > [4] Wang, Tonghan, et al. Roma: Multi-agent reinforcement learning with emergent roles, 2020.

---

> > > > ### Comment · Reviewer_UNCB · 2022-08-08
> > > > **Response to Author Rebuttal**
> > > >
> > > > I thank the authors for the clarification, and these address my questions.

---

### Official Review · Reviewer_LYEL · 2022-07-11

**Rating:** 6
**Confidence:** 4
**Soundness:** 3 good
**Presentation:** 3 good
**Contribution:** 2 fair

**Summary:**

This paper proposes Model-Based Value Decomposition (MBVD), an implicit model-based multi-agent reinforcement learning (MARL) method that integrates the idea of model learning into value decomposition. The authors are enlightened by the ability of long-term vision of humans, which refers to humans’ consideration of future states after several interactions with the environment following their current strategies. Thus the authors let the agents cooperate by obtaining the aggregated latent states in the future. They elaborate on MBVD by discussing the framework, the flowchart, and the learning objectives of the algorithm. The MBVD method is tested in different domains, such as StarCraft II, Google Research Football, and Multi-Agent MuJoCo. The paper clarifies that MBVD exceeds the performance of several baselines in terms of sample efficiency. A derivation of the evidence lower bound (ELBO) and the settings of the environments are included in the appendix.

**Questions:**

In lines 33-34, should the “decentralized partial observable Markov decision process” be “decentralized partially observable Markov decision process” instead?

In line 237, what is the definition of Detach(·)? I can only find the description of what this function can do. References can be added here, but are not necessary.

In lines 142 and 149, the authors mentioned that they used two VAEs as the prior and the posterior for the MBVD framework. Because there have been many variants of the VAE model since it was proposed by Kingma and Welling in 2014, I don’t know which one was used by the authors or if they made a new variant.

From what I understand, when we apply Equation 2 to the experiments in Google Research Football and Multi-Agent MuJoCo, the last error (L_{FA}) is 0 because predictions of the feasible action set are only made in StarCraft II?


**Limitations:**

The authors specified the limitation of the paper is that the value of the rollout horizon was chosen manually. It would be nice if they can elaborate on the limitations, for example, by discussing how other researchers deal with the rollout horizon in their works. It does not seem likely that this paper will cause any potential negative social impact.

**Strengths And Weaknesses:**

Strengths:

This work has good originality. The authors state that the study is the first attempt to apply the model-based reinforcement learning ideas to decentralized partially observable Markov decision process (Dec-POMDP) problems, which is true to the best of my knowledge.
The manuscript is clearly written. Important contents, such as the method itself and the results of the experiments, are presented in a well-organized way. Some more details like the experimental setup are placed in the appendix.

The proof of breaking down the ELBO (Eq. 1) is given in the appendix, which is correct to me and improves the soundness of the submission. The manuscript proves itself to be a complete piece of work rather than a work in progress.

Weaknesses:

In the Related Work section, the authors used a long paragraph to talk about Single-Agent Model-based RL. However, it would be better if the majority of the discussion is on Multi-Agent Model-based RL because the authors’ research is exactly about MARL. There may be relatively fewer papers on multi-agent model-based RL, but the authors can discuss each of them in greater detail.

In the Experiments section, the performance of MBVD is not very impressive in relatively simpler scenarios of StarCraft II. It is hard to decide whether MBVD outperforms other baselines in every hard scenario because only representative scenarios are selected. Also, the ablation studies only show the results from the 2c_vs_64zg map. I am curious to see results from other scenarios for a better generalization.

---

> ### Author Response · Authors · 2022-08-01
> **We thank the reviewer for the inspiring comments. We provide experimental results on other scenarios and discuss how to automatically choose the horizon.**
>
> We thank the reviewer for the inspiring comments.
>
> **Q1**: "What is the definition of Detach(·)?"
>
> **A1**: Detach(·) is a function defined by us, which is consistent with the function of the same name in Pytorch. We consider writing it as StopGrad(·) to make it better understood.
>
> **Q2**: "Which variant of VAE was used by the authors?"
>
> **A2**: We used the vanilla VAE, which was proposed by Kingma and welling in [1].
>
> **Q3**: "The last error ($\mathcal{L}_{\text{FA}}$) is 0 when we apply Equation 2 to the experiments in Google Research Football and Multi-Agent MuJoCo?"
>
> **A3**: Yes. Because only the SMAC environment provides invalid actions of agents, such as attacking a dead enemy unit. We use this information given by the environment to make the agent's choice of actions more reasonable when interacting with the learned environment model. The feasible action set is not available in Google Research Football and Multi-Agent MuJoCo, so we set the loss of the optional auxiliary task $\mathcal{L}\_\text{FA}$ to zero. The experimental results show that MBVD without $\mathcal{L}_{\text{FA}}$ can still perform better than other baselines in Google Research Football and Multi-Agent MuJoCo.
>
> **Q4**: The results from other scenarios in the Experiments section.
>
> **A4**: To make comparisons as fair as possible, we evaluate the performance of all algorithms in the same environmental setting (same version of StarCraft II and same SMAC). We avoid unfair comparisons happened in the original paper of some baselines such as RODE [2],  and show their real performance. The version of StarCraft II used in this paper is SC2.4.6.2.69232 instead of the simpler SC2.4.10, which is the same as in [3]. In this version of StarCraft II almost all algorithms fail to solve the task in *6h_vs_8z*  scenarios.  We provide new experimental results on other SMAC scenarios in Appendix C.1. For ablation studies, we also provide results on several other scenarios in Appendix C.2. Overall, the additional results still conform to our conclusions in the paper.
>
>
>
>  In Appendix D of the updated paper, we discuss how other researchers deal with the rollout horizon in their works.
>
> **References**:
>
> [1] Kingma, Diederik P., and Max Welling. Auto-encoding variational bayes, 2013.
>
> [2] Someone proposed an issue in the official repository of RODE to point out that there is an unfair comparison in the original paper because of their modified SMAC.
>
> [3] Samvelyan, Mikayel, et al. The starcraft multi-agent challenge, 2019.

---

### Author Response · Authors · 2022-08-08
**The author-reviewer discussion period is coming to an end.**

The author-reviewer discussion period ends on Aug 9. We hope our responses have addressed the concerns of reviewers and are happy to engage in further discussion for follow-up concerns.

In this paper, we propose a novel model-based multi-agent reinforcement learning method, MBVD. To the best of our knowledge, this is the first model-based multi-agent algorithm that can be perfectly applied to any Dec-POMDP domain. To verify the generalization of MBVD, we evaluate its performance in the stochastic environments *SMAC* and *Google Research Football* and the deterministic environment *Multi-agent Discrete MuJoCo*. To avoid unfair comparisons in the original paper of some baselines, we carry out all the algorithms in the same environment settings and show their real performance. The experimental results demonstrate that our proposed method MBVD is indeed able to improve the sample efficiency of the original algorithm by considering the imagined states.

---

### Meta-Review · Area_Chair_XKR2 · 2022-09-09

**Recommendation:** Accept
**Confidence:** Certain

**Metareview:**

The reviewers appreciated the paper's originality in its combination of existing components, solid theoretical motivation, clear writing, and evaluation on complex problems. For these reasons, I recommend acceptance.

**Award:**

No

---

### Decision · Program_Chairs · 2022-09-14

Accept